# Data-driven plasma equilibrium forecasting in magnetic fusion tokamak

## Abstract

The most promising approach to achieving nuclear fusion is through tokamaks, which confine plasma using magnetic fields. Understanding the current plasma equilibrium state in tokamaks is critical for effective plasma control. Unlike previous studies, which reconstruct equilibrium from magnetic field information, our work forecasts future equilibrium based on past equilibrium states. Specifically, we formulate the plasma equilibrium prediction task as a video prediction task, a well-explored problem in the machine learning community. This formulation allows us to capture the spatio-temporal dynamics of plasma states and provides a foundation for multimodal modeling of data streams from tokamak operations. Our methodology, incorporating a physics-inspired learning technique for physically reliable predictions, achieved plausible results in forecasting future plasma equilibrium up to 200 ms ahead compared to baselines. This approach holds promise for predicting plasma instabilities and preventing disruptions, marking a significant step towards developing stable fusion reactors.

## 1 Introduction

One way to categorize human history is by examining the main energy sources utilized in each era. From the discovery of fire to the exploitation of fossil fuels, and more recently, to the adoption of nuclear energy and solar power, humanity has continually evolved its primary energy sources. To address climate change and global energy shortages facing humanity, and to enter a new era, developing new clean energy sources has become an urgent mission. Nuclear fusion technology emerges as the most promising candidate for next-generation clean energy and has been studied for decades (Sheffield, 1994; Harms et al., 2000; Freidberg, 2007). It offers the potential for nearly limitless, clean energy production by replicating the processes that power the sun.

Currently, the most viable approach for achieving nuclear fusion is through tokamak devices such as the International Thermonuclear Experimental Reactor (ITER), the largest international collaborative scientific project in the world (Rebut, 1995). Tokamaks use powerful electromagnets to confine high-temperature plasma within a vacuum vessel to induce nuclear fusion reactions. However, numerous scientific and engineering challenges hinder the commercialization of nuclear fusion. In detail, one of the primary obstacles is plasma instability and the resulting plasma disruptions (Schuller, 1995). To predict and prevent these issues, it is essential to understand plasma equilibrium, a state where the magnetic field's confining force on plasma particles balances the force of the particles attempting to diffuse outward. While plasma equilibrium information can be computed by analyzing magnetic field data using an algorithm called equilibrium fitting (EFIT) (Lao et al., 1985), its application is limited due to the difficulty of performing real-time calculations.

To overcome these challenges, the nuclear fusion community has recently begun to actively introduce machine learning (ML) and data-driven approaches (Seo et al., 2021; Degrave et al., 2022; Lao et al., 2022; Char et al., 2023; Joung et al., 2023; Seo et al., 2024; Kim et al., 2024). ML techniques are being applied to tasks such as controlling tokamaks, predicting disruptions, and inferring plasma equilibrium. Along with the aforementioned studies, this work aims to address challenges in the nuclear fusion domain using data-driven methods, specifically by forecasting plasma equilibrium in tokamaks. Unlike the previous study that predicted plasma equilibrium using magnetic field data, we propose predicting future plasma equilibrium based on past plasma equilibrium data. This ap-

proach enables us to address the temporal dynamics of plasma, introducing a novel problem setting in fusion research that goes beyond existing studies focusing on current-state reconstruction.

More precisely, we formulated the problem as a video prediction task by interpreting plasma equilibrium data as a sequence of temporal snapshots. We applied widely known video prediction models to plasma data and introduced methods to incorporate plasma physics knowledge into the model training process. As a result, our model was able to predict plasma equilibrium up to approximately 200 ms into the future without explicit tokamak control information. This is considered a sufficiently long-term prediction, as plasma physics typically deals with phenomena occurring on timescales of a few to tens of milliseconds inside a tokamak.

Our research introduces a novel data-driven approach to nuclear fusion, an under-explored field within the ML community. By tackling complex spatio-temporal plasma equilibrium data rather than the scalar features used in previous studies (Wai et al., 2022; Degrave et al., 2022; Wan et al., 2023; Char et al., 2023), we demonstrate the potential of ML in fusion science. In this manner, our work can serve as a foundation for developing multimodal models in fusion research, paving the way for the simultaneous analysis of diverse fusion data types. Our findings, based on data from Korea Superconducting Tokamak Advanced Research (KSTAR) (Lee et al., 2001), are applicable to ITER, given the structural similarities between these tokamaks. Finally, current research efforts are focused on developing a digital twin of KSTAR and conducting virtual fusion experiments (Kwon et al., 2022). Our study has the potential to significantly advance the ultimate goal of comprehensive virtual fusion experimentation.

The contributions of our work can be summarized as follows:

- We formulated the plasma equilibrium forecasting problem as a video prediction task, introducing this under-explored problem to both nuclear fusion and ML communities.
- We introduced a physics-based loss function inspired by plasma equilibrium equations and demonstrated its effectiveness.
- We evaluated our proposed model against widely-used baseline methods across various domains and interpreted the results from the plasma physics perspective.

## 2 PRELIMINARY

This section provides a brief introduction to the background knowledge necessary to understand our work. For a deeper understanding of nuclear fusion or how ML can be applied in the field of fusion, readers may refer to Hutchinson (2002), Ikeda (2007), Freidberg (2007), Wesson & Campbell (2011), McCracken & Stott (2013), and Spangher et al. (2024).

### 2.1 NUCLEAR FUSION AND TOKAMAK

Nuclear fusion is a physical process in which two light atomic nuclei combine into a single heavier nucleus under high temperature and pressure, simultaneously releasing energy. This process results in plasma, known as the fourth state of matter, where electrons are separated from atomic nuclei and can move freely. Einstein's famous equation $E = mc^2$ can be used to calculate the amount of energy produced during nuclear fusion, showing that even a small amount of fuel can generate an enormous amount of energy. For example, to produce the same energy as 125 kg of deuterium and tritium fuel, approximately 2.7 million tons of coal are needed [1].

In fact, nuclear fusion reactions occur in the sun, and its brightness is due to these reactions. While nuclear fusion in the sun occurs at approximately 15 million Kelvin due to immense gravitational pressure, achieving fusion on Earth requires temperatures of 100–150 million Kelvin because we cannot replicate the sun's gravitational conditions. However, building a vessel or material that can directly contact plasma at such high temperatures is practically infeasible. Therefore, researchers have devised a method using magnetic fields to confine the plasma in a vacuum state, preventing it from contacting the vessel walls. This concept is implemented in devices known as tokamaks, which are donut-shaped machines. The tokamak consists of various components, namely multiple electromagnets for confining and controlling plasma, heating systems such as neutral beam injection

---

[1] https://www.iter.org/sci/FusionFuels

(NBI) and electron cyclotron resonance (ECR) heating systems, and multiple types of sensors for monitoring the tokamak's operational status.

Representative tokamak devices, including KSTAR (Lee et al., 2001) in South Korea, DIII-D (Luxon, 2002) in the United States, EAST (the EAST et al., 2009) in China, and JET (Jacquinot et al., 1999) in Europe, are contributing crucial insights into plasma physics as well as the development and operation of ITER (Rebut, 1995). Among these, KSTAR and EAST, which leverage superconducting electromagnets, can operate for significantly longer periods compared to other tokamaks that do not use superconductors. This key difference provides a distinct advantage, enabling long-term plasma experiments and detailed investigations. Our work utilizes real experimental data from KSTAR; consequently, our findings have potential applications in the prediction and analysis of long-duration tokamak operations.

## 2.2 PLASMA EQUILIBRIUM

The most crucial information for controlling and analyzing plasma inside a tokamak is plasma equilibrium. Plasma equilibrium refers to the balance between the magnetic forces confining the plasma and the outward pressure of the plasma itself. The Grad-Shafranov (GS) equation (Grad & Rubin, 1958; Shafranov, 1966), a partial differential equation describing this equilibrium state, illustrates the spatial distribution of magnetic fields under conditions satisfying plasma equilibrium:

$$\Delta^*\psi = -\mu_0 R J_\phi = -\mu_0 R^2 \frac{dp(\psi)}{d\psi} - F(\psi)\frac{dF(\psi)}{d\psi}. \tag{1}$$

In this equation, $\psi = \psi(R, Z)$ represents the poloidal magnetic flux function dependent on coordinates $R$ and $Z$, defined within the cylindrical coordinate system of KSTAR. The term $\mu_0$ denotes the permeability of free space, while $p(\psi)$ represents the plasma pressure and $F(\psi)$ is related to the toroidal magnetic field, both being functions of $\psi$. $\Delta^*\psi$ is a two-dimensional non-linear partial derivative of $\psi$, defined as $\Delta^* = R\frac{\partial}{\partial R}\frac{1}{R}\frac{\partial}{\partial R} + \frac{\partial^2}{\partial Z^2}$.

Based on the $R$, $Z$, and $J_\phi$, which is the toroidal component of the plasma current density, we can obtain the solution $\psi$ (and $\Delta^*\psi$) of the GS equation. This information serves as an essential key to understanding the state of the plasma for controlling and analyzing the tokamak. However, it is challenging to determine the solution of the GS equation, and the EFIT algorithm (Lao et al., 1985) was developed to effectively find the solution. In brief, EFIT employs an iterative calculation process to find the solution based on given initial conditions. Therefore, it has limitations in scenarios requiring real-time application such as tokamak control due to its heavy computational time. Consequently, the real-time EFIT (rt-EFIT) algorithm (Ferron et al., 1998; Moret et al., 2015; Huang et al., 2017) has been proposed to reduce computational time by using the solutions at the previous time step as initial values, but this approach is also limited due to lower accuracy.

The reconstruction of plasma equilibrium through EFIT calculations provides abundant information for both controlling and understanding plasma behavior. Primarily, the key output $\psi$ is utilized to determine the last closed flux surface (LCFS). The LCFS, which delineates the boundary of the confined plasma, is a crucial parameter in tokamak operations. By accurately identifying the LCFS, researchers can enhance plasma stability, optimize tokamak operational methods, and gain deeper insights into plasma dynamics. Furthermore, the results from the EFIT calculations can be combined with various experimental data collected from tokamak diagnostics. This allows for the derivation of important plasma parameters, including temperature profiles, density distributions, and magnetic pressure, which are crucial for optimizing plasma stability, confinement efficiency, and fusion reaction rates.

## 2.3 DATA-DRIVEN APPROACHES IN FUSION DOMAIN

Due to the computational cost of the EFIT algorithm, data-driven approaches have been recently proposed as alternatives to predict plasma equilibrium faster and with comparable accuracy to the EFIT algorithm (Joung et al., 2019; Wai et al., 2022; Lao et al., 2022; Joung et al., 2023; Lu et al., 2023). GS-DeepNet serves as a notable example (Joung et al., 2023). This model predicts the current plasma equilibrium in real time with the current magnetic field information given as input. Notably,

GS-DeepNet introduced an unsupervised learning mechanism without requiring ground truth EFIT data. Our method, in contrast, predicts the current or future equilibrium based on past equilibrium information without magnetic field data, and can potentially collaborate with the aforementioned methods by bootstrapping from the equilibria they reconstruct.

Beyond the plasma equilibrium prediction, data-driven techniques are increasingly being applied to various challenges in the field of nuclear fusion. For example, Seo et al. (2021), Degrave et al. (2022), and Char et al. (2023) demonstrated the potential of ML in tokamak control; moreover, Seo et al. (2024) and Kim et al. (2024) showed that ML-based control can help avoid plasma instabilities, which are a major challenge in tokamak operation. While most studies have utilized a number of simple and raw features from the tokamak as inputs, our method distinguishes itself by leveraging spatio-temporal information, such as $\psi$ and $\Delta^*\psi$ from the EFIT calculation. This approach allows us to capture complex plasma dynamics and their evolution over time, providing a framework that can be extended to support diverse types of data with complex structures generated from tokamaks. Our study, along with existing research, establishes a foundation for the development of multimodal models capable of integrating and understanding the enormous volume of data from tokamaks.

## 3 METHOD

In this section, we define our problem setup and introduce our proposed approach. Our key idea is to interpret the plasma equilibrium prediction task as a video prediction task in the ML domain. We present the advantages of this approach, as well as propose a technique inspired by plasma physics.

### 3.1 FORMULATING PLASMA EQUILIBRIUM PREDICTION AS VIDEO PREDICTION

Video prediction is a task of predicting future frames based on the given past frames. Formally, given $T$ past frames $X_{t-T:t-1} = \{x_i\}_{t-T}^{t-1}$ at time $t$, the objective of this task is to obtain $T'$ future frames $Y_{t:t+T'-1} = \{x_i\}_t^{t+T'-1}$, where $x_i$ represents an image of dimensions $C \times H \times W$, with $C$ channels and spatial resolution of $H \times W$. The EFIT data from fusion experiments have similar features to video data, allowing us to approach fusion plasma prediction as a video prediction problem. In this context, each EFIT frame $x_i$ can be considered as a two-dimensional image where channels correspond to $\psi$ and $\Delta^*\psi$.

This video-based approach offers several advantages over the traditional pixel-level approach, which treated $\psi$ and $\Delta^*\psi$ values at each point as separate data. By simultaneously utilizing $\psi$ and $\Delta^*\psi$ information from multiple points, we can expect more accurate predictions. For instance, this comprehensive approach enables the model to identify complex spatial patterns and relationships that traditional pixel-level methods could not capture. Furthermore, this framework has the potential to be extended to a multimodal model in the future by employing additional spatio-temporal data collected from tokamaks. Moreover, by processing spatial and temporal information concurrently, this method offers an opportunity to analyze plasma dynamics from a spatio-temporal perspective. This is particularly significant because the aforementioned GS equation, which is currently the primary method for reconstructing plasma equilibrium, describes only spatial changes but not temporal evolution. If the model based on video prediction approach provides us with significant results regarding plasma's temporal dynamics, it could lead to novel scientific insights into plasma behavior.

This intersection of physics and ML is promising (Chen et al., 2022), as ML researchers' expertise in video prediction can offer fresh perspectives unconstrained by traditional physical knowledge. To this end, we explore the application of SimVP (Gao et al., 2022), a widely adopted and versatile video prediction model architecture. SimVP processes input frames by employing spatial downsampling, transforming spatio-temporal features through inception modules, and decoding them to generate predictions at the original resolution. Despite the existence of various video prediction models, we selected this one for its straightforward and versatile nature in video prediction tasks. Specifically, SimVP does not make assumptions about the data type, making it an ideal candidate for the plasma equilibrium prediction task. This flexibility is crucial for our future plans to develop multimodal models capable of integrating diverse data from tokamak operations.

## 3.2 GRAD-SHAFRANOV CONSTRAINT LOSS

The $\psi$ and $\Delta^*\psi$ data used as input and output in our study are derived from the EFIT algorithm, which is based on the GS equation (Equation 1 in Section 2.2). Essentially, these data inherently satisfy the GS equation, implying that any accurate prediction from our model should also follow this fundamental constraint. To ensure this, we introduce a GS-constraint loss to train the model in addition to the mean absolute error (MAE) on $\psi$ and $\Delta^*\psi$. The GS-constraint loss helps to maintain the physical consistency of our predictions within the theoretical foundation of plasma equilibrium.

In detail, we apply $\Delta^*$ operator, defined as $\Delta^* = R\frac{\partial}{\partial R}\frac{1}{R}\frac{\partial}{\partial R} + \frac{\partial^2}{\partial Z^2}$, to the model output $\psi_{pred}$ to calculate $\Delta^*\psi_{est}$. Subsequently, we compare it with the model output $\Delta^*\psi_{pred}$ to compute the MAE, which serves as our GS-constraint loss as follows:

$$\mathcal{L}_{GS} = \mathbb{E}\left\|\Delta^*\psi_{est} - \Delta^*\psi_{pred}\right\|_1 = \mathbb{E}\left\|D_{\Delta^*}(\psi_{pred}) - \Delta^*\psi_{pred}\right\|_1. \tag{2}$$

Here, $D_{\Delta^*}(\psi_{pred})$ represents the application of the $\Delta^*$ operator to $\psi_{pred}$ using finite difference approximations. Since $\psi$ and $\Delta^*\psi$ are defined on a fixed grid over $R$ and $Z$, and since the coordinates $R$ and $Z$ are fixed in our data representation, we cannot use automatic differentiation to compute spatial derivatives in this case. Alternatively, following Ren et al. (2022), we approximated the $\Delta^*$ operator by employing a finite difference method, shifting $\psi$ horizontally and vertically. Although this approximation technique is numerically less accurate than actual differentiation, it presents the most feasible solution for our problem. The approximation error (reported in Table 1 using $^\dagger$ symbol) in the GS-constraint loss term could lead to inaccurate predictions. Therefore, we scaled the GS-constraint loss by a coefficient less than one and added it to the pixel MAE loss in order to mitigate numerical errors: $\mathcal{L}_{Total} = \mathcal{L}_{MAE} + \alpha\mathcal{L}_{GS}$, where $\alpha = 0.01$.

Meanwhile, one might propose using the leftmost and rightmost terms of Equation 1 for constraining the model predictions. This approach could allow for the natural calculation of derivative values through automatic differentiation. However, this approach faces practical limitations due to the ambiguity in the GS equation, which does not provide information about the exact forms of $p(\psi)$ and $F(\psi)$. Specifically, although we understand the physical meaning of $p$ and $F$, we cannot precisely determine their functional forms or characteristics as functions of $\psi$. In practice, the spatial distribution of current ($FF'$ term) in the GS equation is typically approximated using low-order functions, such as first to third-order polynomials. Consequently, we opted for the previously described loss based on finite differences in a conservative manner. This conservative choice was made to ensure both numerical stability during training and physical fidelity of the model, as the alternative method could potentially lead to ambiguous physical interpretations and unstable training results. While this limitation could be addressed if additional conditions or data for $p$ and $F$ were available, such considerations remain a topic for future research [2].

## 4 EXPERIMENTS

This section presents the experiments and results that demonstrate the effectiveness of our approach. It begins by introducing the dataset used for our experiments, along with the model training details. We then introduce the evaluation metrics, which are adaptations of commonly used metrics in ML problems, tailored for the plasma equilibrium prediction context. Finally, we conduct a comprehensive analysis through both quantitative and qualitative assessments of the experimental results.

### 4.1 KSTAR DATASET

We evaluated our method using experimental data from KSTAR. Our dataset comprises 789 KSTAR discharges collected from the 2017 and 2018 campaigns, split into training and test sets with an 8:2 ratio. A single data instance includes offline EFIT calculation results, plasma current ($I_p$) information, and heating information (NBI and ECR). The EFIT data, which are the primary data used in our study, consist of a two-channel video ($\psi$ and $\Delta^*\psi$) for each shot, calculated at 50 ms intervals.

---

[2]In our problem setup, $p(\psi)$ and $F(\psi)$ can be implemented by stacking $1 \times 1$ convolutional layers.

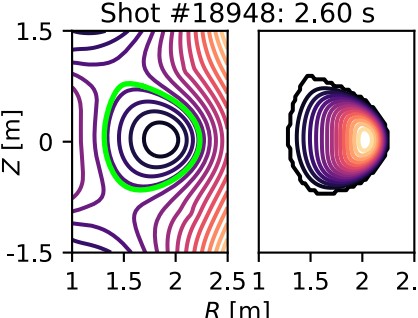 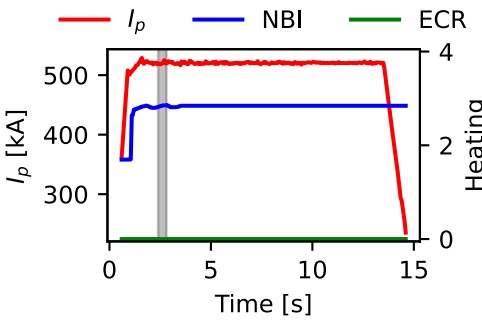

Figure 1: Visualization of example KSTAR EFIT data used in this work. The left plot shows $\psi$ and $\Delta^*\psi$. The horizontal and vertical axes represent the $R$ and $Z$ coordinates of KSTAR, respectively, measured in meters. The original $\psi$ and $\Delta^*\psi$ data are displayed as heatmaps, while the researchers usually present them as contour lines. The LCFS, highlighted in lime, indicates the current confinement state and configuration of the plasma. The right plot illustrates plasma current information along with heating information. The plasma current, shown in red on the left y-axis, represents the overall operational status of the shot. Heating information is displayed on the right y-axis, with a blue line for NBI and a green line for ECR. The gray-shaded region represents the time interval for which the EFIT data are depicted in the upper plots.

Figure 1 illustrates the EFIT data from KSTAR discharge shot #18948 at 2.60 seconds. Each frame consists of a two-channel heatmap representing $\psi$ and $\Delta^*\psi$, which satisfy the GS equation. The original EFIT data was calculated at a $65 \times 65$ resolution, which is specifically calibrated resolution for KSTAR operations, but we cropped one pixel from the right and bottom, resulting in a $64 \times 64$ resolution for convenience. In addition, we applied min-max normalization to $\psi$ and $\Delta^*\psi$ based on their statistics. The preprocessed EFIT data are visualized using contour lines, as shown in Figure 1. The most important feature for fusion researchers is the LCFS, drawn in a lime-colored line, which represents the plasma boundary. The LCFS is one of the most essential elements in plasma control and analysis, requiring precise estimation. To determine the LCFS, we generated 80 contour lines and selected the outermost closed line as the LCFS.

In addition to the EFIT data, we employed plasma current and heating information for each shot. Although plasma current information was not used in training, it was utilized in the analysis process because it provides insight into the overall operational status of the shot. The heating information contains both NBI and ECR data. For analysis purposes, we used the raw heating data, but downsampled it to match the EFIT sampling rate during training. This plasma current and heating information shows the operational status of the tokamak for each shot. For instance, examining the right plot in Figure 1, which illustrates discharge # 18948, we can observe the overall operational phases: a ramp-up stage until around 2 seconds, followed by a flat-top, and a ramp-down starting near 13 seconds. Furthermore, this experiment did not utilize ECR heating, but employed three NBI devices for plasma heating. Preprocessing of the NBI and ECR data was minimal due to their less significant numerical deviations. We only replaced negative values in the NBI and ECR with zeros, and there was no further normalization. When using the heating information in training, we replicated the scalar values of heating information to match the shape of $\psi$ and $\Delta^*\psi$. Then, these replicated values were concatenated with $\psi$ and $\Delta^*\psi$ to form a four-channel input data.

## 4.2 BASELINE AND EVALUATION METRIC

**Evaluation metric** The basic evaluation was conducted using the pixel-level MAE, a common metric in video prediction tasks. In our work, we computed MAE for specific regions of interest in the $\psi$ and $\Delta^*\psi$, rather than considering the entire region. The first area is the central region. This area was prioritized because the prediction accuracy in the outer regions, where plasma does not exist, is relatively less important. Second, we assessed the divertor region. The divertor is a crucial component of the tokamak, responsible for extracting high-temperature exhaust and residues from the plasma. Accurate prediction of plasma boundary in the divertor is essential for sustaining long-period operations and preventing damage to the tokamak. In detail, the plasma boundary al-

lows us to predict and evaluate the accuracy of key features such as the X-point (the intersection of magnetic flux surfaces) and the divertor strike points where plasma and residues make contact with the divertor. However, predicting these features accurately requires centimeter to millimeter-scale precision (Eldon et al., 2020), which presents a challenge for the spatial resolution of our current EFIT data. In this context, we extended our evaluation to assess the accuracy of the LCFS determined from the predicted $\psi$. As a potential alternative evaluation metric, we experimented with intersection over union (IoU) and mean IoU (mIoU) to compare the similarity between the ground truth and predicted LCFS shapes. The detailed analysis of these metrics and their limitations are discussed in Section B.

**Baseline**  Since our proposed method differs from existing research in both approach and results, it is challenging in making direct and fair comparisons. To address this, we selected several baseline methods that are widely adopted across different domains. First, we employed a time-lagged baseline, widely recognized as a simple yet robust benchmark in time series forecasting problems. This baseline outputs ground truth values from our EFIT dataset, delayed by a single time step. In detail, we applied a 50 ms time lag, which represents a significantly long period in the context of plasma dynamics. For example, plasma dynamics is typically analyzed with a time resolution of less than 10 ms, and tokamak control systems require a response time on the order of a few ms. We also presented a constant prediction baseline that assumes that the most recently observed plasma state persists unchanged. Next, we conducted experiments using ConvLSTM (Shi et al., 2015), PredRNN (Wang et al., 2017), and PredRNN-v2 (Wang et al., 2023), which are fundamental video prediction models. Finally, we compared our results with dynamic mode decomposition (DMD) (Schmid, 2010), a widely-used linear model for predicting spatio-temporal dynamics, and Chronos (Ansari et al., 2024), a general-purpose pretrained time series forecasting model.

### 4.3 TRAINING AND IMPLEMENTATION DETAILS

In our experiments, we used the implementation of model architectures from the `OpenSTL` library (Tan et al., 2022; 2023), while developing training and evaluation code from scratch to fit our problem setup. For hyperparameters, SimVP was configured with $N_S = 4$ and $N_T = 8$ layers, with hidden dimensions of $d_S = 64$ and $d_T = 256$, respectively. The ConvLSTM model consisted of four layers, each with a hidden dimension of 96. PredRNN and PredRNN-v2 were also configured with four layers, but each utilized a hidden dimension of 128. To simplify our experiments, we set the input and output frame lengths to be equal, in other words, $T = T'$. These sequences contain one, four, or eight frames, representing time spans of 50 ms, 200 ms, and 400 ms, respectively.

The models shared the following training hyperparameters: We utilized the AdamW optimizer (Loshchilov & Hutter, 2019) with a learning rate of $10^{-4}$, weight decay of 0.05, and gradient clipping of 1. The dataset was sampled every four timesteps, and the batch size was set to 128. The number of training epochs was $1,500$, and we applied an exponential moving average update with $\beta = 0.9999$. To ensure statistical robustness, all experiments were repeated with three random seeds, and we reported the average values. Each experiment was conducted on a single NVIDIA H100 GPU and completed within 48 hours.

Predictions with DMD were generated autoregressively from the last observation after fitting it with $T$ past frames. Since DMD inherently operates on 1D vectors, we tested two approaches: (1) flattening the 2D EFIT data into 1D vectors and (2) performing predictions on a per-pixel basis. For the flattening approach, we considered both joint prediction of $\psi$ and $\Delta^*\psi$ (*i.e.*, predicting a vector of dimension $64 \times 64 \times 2 = 8,192$) and separate prediction (*i.e.*, predicting two vectors, each of dimension $64 \times 64 = 4,096$). All DMD experiments were conducted using the `PyDMD` library (Demo et al., 2018). As for the Chronos model, we employed `Chronos-t5-base`, which has a comparable number of parameters to SimVP, and assessed its zero-shot accuracy without finetuning. To enhance reliability, the final predictions of the Chronos model were obtained by averaging the outcomes of 15 repeated samplings for each data point.

### 4.4 RESULTS AND ANALYSIS

We evaluated the efficacy of the GS-constraint loss and conducted a comprehensive analysis of the prediction accuracy under various scenarios, varying both the types of input data and the prediction length. Table 1 summarizes our experimental results. Among the baselines presented in the first and

Table 1: Qualitative evaluation of plasma equilibrium prediction.

| Method | $\psi$ MAE ($\downarrow$) | | $\Delta^*\psi$ MAE ($\downarrow$) | | GS |
| --- | --- | --- | --- | --- | --- |
| | Center | Divertor | Center | Divertor | const. ($\downarrow$) |
| Baseline (50 ms delay) | 0.00165 | 0.00161 | 0.01493 | 0.00742 | 0.04679$^\dagger$ |
| Baseline (Constant) | 0.00676 | 0.00687 | 0.02946 | 0.01533 | 0.04679$^\dagger$ |
| DMD (200 ms) | 0.93483 | 0.89436 | 14.4225 | 9.37053 | 5.09277 |
| DMD (200 ms) $\psi$ only | 0.98264 | 1.26488 | - | - | - |
| DMD (200 ms) pixel-level | 0.01053 | 0.01028 | 0.03441 | 0.01965 | 0.03163 |
| Chronos-base (200 ms) | 0.01758 | 0.01756 | 0.06306 | 0.04728 | 47.0440 |
| ConvLSTM (200 ms) | 0.00385 | 0.00463 | 0.04160 | 0.02701 | 0.34480 |
| PredRNN (200 ms) | 0.00287 | 0.00355 | 0.05799 | 0.03089 | 1.54382 |
| PredRNNv2 (200 ms) | 0.00603 | 0.00720 | 0.06159 | 0.03474 | 2.01905 |
| SimVP (400 ms) | 0.00175 | 0.00178 | 0.02434 | 0.01365 | 0.38020 |
| SimVP (400 ms) + $\mathcal{L}_{GS}$ | 0.00165 | 0.00172 | 0.02396 | 0.01347 | 0.21892 |
| SimVP (200 ms) | 0.00128 | 0.00124 | 0.01825 | 0.00996 | 0.36242 |
| SimVP (200 ms) + $\mathcal{L}_{GS}$ | 0.00123 | 0.00125 | 0.02035 | 0.01108 | 0.21476 |
| SimVP (200 ms) $\psi$ only | 0.00122 | 0.00119 | - | - | - |
| SimVP (50 ms) | 0.00116 | 0.00109 | 0.01850 | 0.01007 | 0.47197 |
| SimVP (50 ms) + $\mathcal{L}_{GS}$ | 0.00106 | 0.00103 | 0.01945 | 0.01050 | 0.22314 |
| SimVP (200 ms) + $\mathcal{L}_{GS}$ + Heat. | 0.00181 | 0.00194 | 0.03291 | 0.01823 | 0.23188 |
| SimVP (200 ms) + $\mathcal{L}_{GS}$ + Rev. | 0.00132 | 0.00136 | 0.02289 | 0.01229 | 0.22324 |
| SimVP (50 ms) + $\mathcal{L}_{GS}$ + Heat. | 0.00172 | 0.00176 | 0.02330 | 0.01296 | 0.22133 |

second groups of the table, the time-lagged baseline demonstrated the lowest MAEs. This result is primarily due to the averaged metrics being reported across all timesteps. Notably, as illustrated in Figure 4 in Section A, the 50 ms ahead prediction accuracy of the pixel-level DMD model is comparable to that of SimVP. Further details can be found in A.

**GS-constraint loss** The third group in Table 1 demonstrates the changes in MAE when the GS-constraint loss was introduced. The models using the GS-constraint loss generally showed lower MAEs on $\psi$ and the predicted $\psi$ and $\Delta^*\psi$ more closely satisfied the GS equation. However, we observed performance degradation in $\Delta^*\psi$ prediction in the case of the 50 ms and 200 ms models. This trade-off is acceptable considering that $\psi$ prediction is relatively more important than $\Delta^*\psi$ prediction in the nuclear fusion domain. We also conducted experiments predicting only $\psi$, which resulted in improved prediction accuracy. Nevertheless, it should be noted that while the accuracy of $\psi$ predictions may increase, this does not necessarily guarantee their physical validity; thus, further investigation is required to determine whether using only $\psi$ is practically acceptable.

**Input configuration and Prediction length** Prior to our experiments, we hypothesized that incorporating heating information, which partially represents control signals of the tokamak, would enhance the prediction accuracy of the models. However, contrary to our expectations, Table 1 and Figure 2 (a) demonstrate that the inclusion of heating data did not bring improvement, or led to a decrease in accuracy in some cases. These unexpected results suggest that developing a multi-modal model capable of integrating multiple data types from a tokamak requires more than simply adding extra input channels. It indicates a need for architectural improvements in the model design to properly leverage these additional data sources.

Alternatively, we can hypothesize that the current heating information may be insufficient for accurate forecasting of $\psi$ and $\Delta^*\psi$. In fact, heating in a tokamak is not limited to external devices such as ECR and NBI. There is also Ohmic heating, where the plasma is heated by its own electrical resistance. If Ohmic heating contributes more significantly to the future plasma equilibrium than the ECR and NBI, this could explain the current performance degradation when including heating data.

Subsequently, we examined the impact of prediction length on the models. In typical time series forecasting tasks, shorter prediction lengths generally yield lower error rates. As evidenced in Fig-

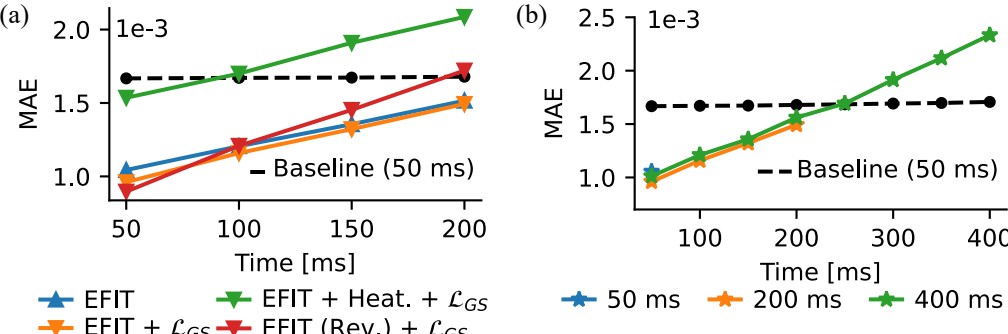

Figure 2: MAEs of $\psi$ predictions across various model and input configurations. These two plots show MAE values for $\psi$ predictions in the center region at different future time points. (a) All models, except the one using heating information, more accurately predict $\psi$ up to 200 ms into the future compared to the 50 ms lagged baseline. (b) Models with input and output lengths of 200 ms and 400 ms exhibit higher prediction accuracy at 50 ms compared to the model with 50 ms input and output lengths.

ure 2 (b), our results align with this trend, showing that models predicting 200 ms into the future achieve lower MAEs compared to those predicting 400 ms. Interestingly, we observed that the model predicting only 50 ms ahead demonstrated higher errors than the other two models, contradicting our initial intuition.

This unexpected finding raises the possibility that plasma states from the more distant past (beyond 50 ms) might have a potential influence on future plasma equilibrium, and this hypothesis requires further rigorous investigation to confirm. If substantiated through future study, this observation might offer a new perspective on plasma dynamics from a temporal viewpoint. Such a perspective could potentially complement the spatial description provided by the GS equation, which illustrates plasma equilibrium in space. However, it is crucial to emphasize that these are preliminary interpretations. Extensive experimental and theoretical investigations would be necessary to validate these hypotheses for understanding plasma dynamics.

In addition, we recognize that our current problem setting may not always require long-term equilibrium predictions. In fact, for many practical applications, short-term predictions are often sufficient and more relevant. To enhance prediction accuracy for the nearest future (50 ms ahead), we experimented with a counterintuitive technique of reversing the input sequence. The rationale behind this approach stems from the architecture of SimVP, where convolutional layers update the temporal information of input features. The features of the earliest input frame contribute most significantly to calculating the features of the nearest future frame in the output. By reversing the input sequence, we enable the most recent historical information to be utilized in computing the nearest future information. Figure 2 (a) shows that the model with reversed input performs best at 50 ms, validating the effectiveness of this heuristic idea. This technique could potentially be further refined by applying loss only to the nearest future frames.

**Qualitative analysis** Lastly, we analyzed the actual prediction results by visualizing the model outputs. Figure 3 shows the prediction results for discharge shot #20279 from 9.35 s to 9.50 s. This corresponds to the ramp-down phase of the experiment. The ramp-down phase involves reducing energy after maintaining plasma in a stable confinement state (*i.e.*, the flat-top phase). In other words, it is the process of terminating the fusion reaction and experiment. We focused our analysis on this phase because, during ramp-down, confinement gradually weakens, leading to plasma instabilities and an increased risk of disruptions, requiring careful control.

The results show that the predicted plasma boundary closely matches the ground truth. However, prediction accuracy decreases near the X-point region, suggesting that further refinement is necessary to achieve the level of precision required for plasma shaping and divertor control. Despite this limitation, it is noteworthy that we achieved this level of prediction using only past equilibrium states. This suggests that by enhancing forecasting accuracy through integrating these data and architectural improvements, the model could be practically used in various scenarios.

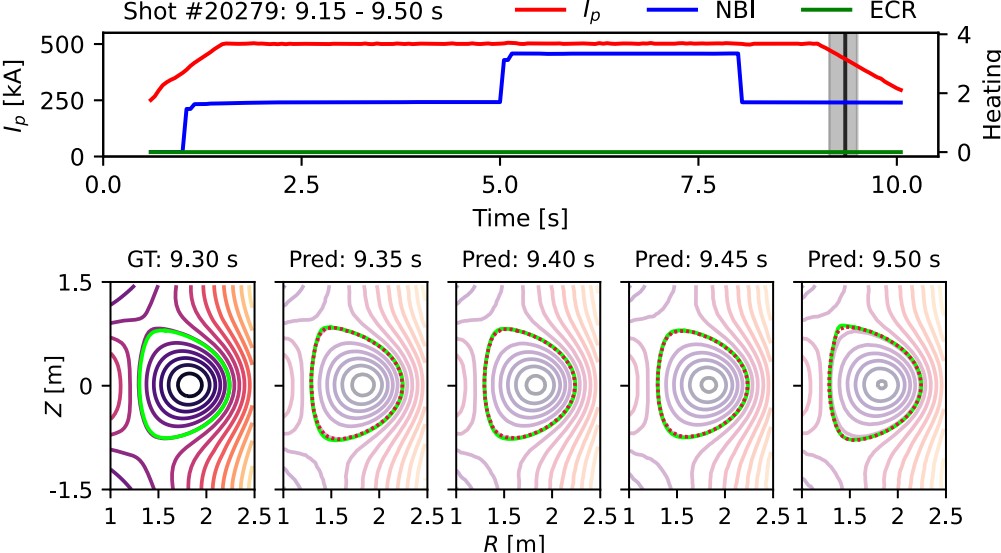

Figure 3: Visualization of model predictions at Shot #20279. The model uses $\psi$ and $\Delta^*\psi$ values from 9.15 s to 9.30 s as input to predict the corresponding values from 9.35 s to 9.50 s. The top plot illustrates the overall operational status of the shot, with input and output intervals shaded in gray and separated by a black vertical line. In the bottom row, the leftmost plot displays the last frame of the input $\psi$, while the subsequent plots show the predicted results in chronological order. In each $\psi$ plot, the ground truth LCFS is depicted by a lime-colored line, and the LCFS determined from the model output is shown as a red dotted line.

## 5 DISCUSSION AND CONCLUSION

In this paper, we introduced a novel problem of forecasting future plasma equilibrium in tokamaks based on past states. We formulated this as a video prediction problem and conducted experiments across various methods and scenarios. Our results showed that the models can predict the poloidal magnetic flux ($\psi$) up to 200 ms into the future more accurately than the baselines. In addition, we found that introducing a loss function based on the GS equation improved the predictions of $\psi$ and resulted in physically reliable forecasting. However, we observed an unexpected accuracy degradation when integrating partial tokamak control signals, *i.e.*, NBI and ECR heating data, suggesting the need for further investigation to effectively leverage additional data from the tokamak. We conclude by discussing the potential applications of our results in the fusion domain and future directions.

**Outlook** In recent years, the nuclear fusion community has shown increasing interest in developing digital twins of tokamaks (Kwon et al., 2022; Iglesias et al., 2017). Beyond simple 3D models, these digital twins serve as tools for virtual fusion experiments that integrate various experimental data and simulation results. This approach offers two significant advantages: it reduces the time and energy costs required for each discharge, and it can minimize the risk of damage in the tokamak that may occur during experiments. A notable example is the Virtual KSTAR (Kwon et al., 2022), a digital twin of KSTAR. The techniques developed in this project are expected to be applied in the future development of digital twins for ITER and K-DEMO (Cho et al., 2022).

Typically, fusion experiments are conducted based on detailed control scenarios, and virtual experiments follow a similar approach; The system should be capable of generating simulated plasma discharge results based on input scenarios describing the operational plan. To achieve this, a multimodal model is required to efficiently utilize not only current EFIT data but also various data from the tokamak, including heating information and magnetic field data (*i.e.*, control information). In this context, our proposed method for predicting plasma equilibrium can be extended to serve as a foundation for fusion multimodal models. We believe that our results can potentially contribute to predicting plasma instabilities, preventing disruptions, and gaining novel insights into plasma dynamics, thereby taking a significant step towards realizing data-driven stable fusion reactors.

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

## A ADDITIONAL QUANTITATIVE RESULTS

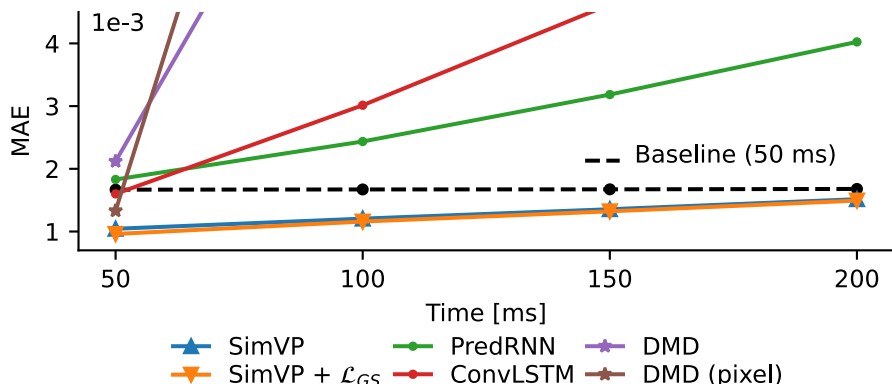

Figure 4: MAEs of $\psi$ predictions across various methods. This plot presents MAE values for $\psi$ predictions in the center region at different future time points. All methods, except SimVP, had lower accuracy than the 50 ms lagged baseline. At 50 ms ahead, DMD with pixel-level predictions performed comparably to SimVP. The results of Chronos and PredRNNv2, which showed relatively larger errors compared to other models, were omitted.

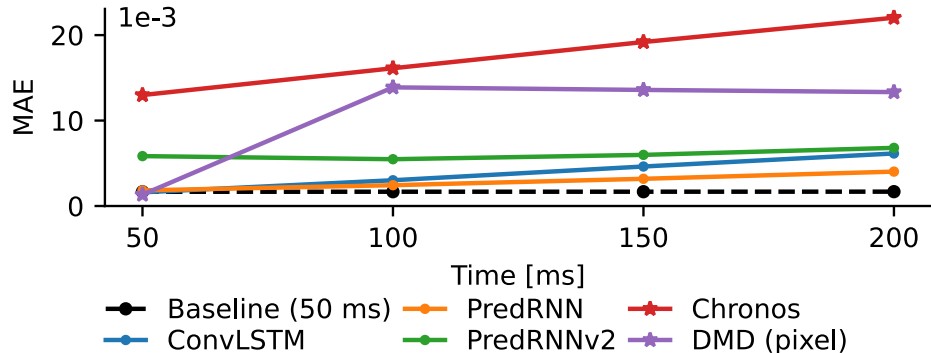

Figure 5: MAEs of $\psi$ predictions in the center region at different future time points. This plot shows the models (Chronos and PredRNNv2) and y-axis values omitted in Figure 4. The MAE values of the DMD model, which reach up to 3.59, are excluded due to their excessively high magnitude. Unlike other methods, the pixel-level DMD model exhibits a rapid increase in error at 100 ms.

Table 1 illustrates that the models including DMD, Chronos, ConvLSTM, PredRNN, and PredRNNv2 demonstrated higher errors compared to the time-lagged baseline. However, Figure 4 demonstrates that these models did not fail entirely in predictions across all timesteps. For predictions 50 ms ahead, all methods except PredRNNv2 and Chronos achieved accuracy comparable to the baseline. Notably, DMD with pixel-level predictions performed similarly to SimVP.

Beyond 100 ms, the prediction errors of the DMD models increased sharply, exhibiting the steepest rise among the comparative models. This may be attributed to numerical instability during the fitting process of the DMD models, particularly when simultaneously predicting $\psi$ and $\Delta^*\psi$. Evidence includes warning messages from the `PyDMD` library and the observation that the MAE at 200 ms reaches 3.59, which is approximately 1,700 times larger than the error at 50 ms.

Interestingly, even the pixel-level DMD model, which involves far fewer variables, exhibits a significant increase in error as depicted in Figure 5. This suggests that additional factors contribute to the observed performance drop. One possible explanation is that accurately predicting the temporal dynamics of plasma requires accounting for spatial information. Furthermore, the non-linear nature of plasma dynamics may limit the ability of linear models such as DMD to predict long-term behavior beyond 100 ms. However, these hypotheses cannot be conclusively verified with the current experimental results, highlighting the need for further investigation.

## B    DETAILS AND LIMITATIONS OF IoU-BASED LCFS EVALUATION

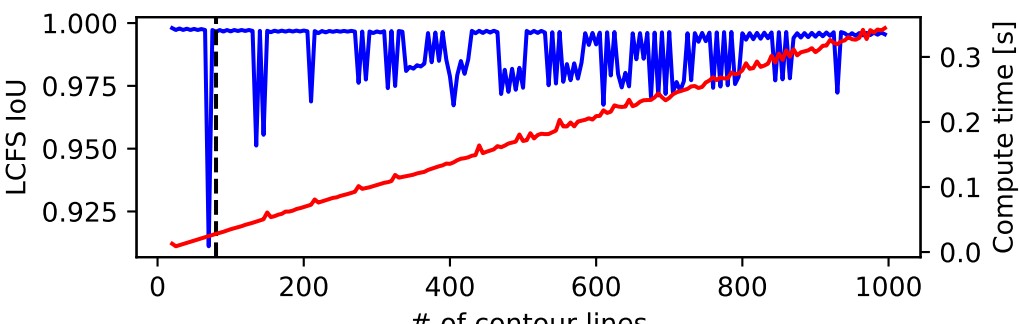

Figure 6:   IoU measurements of the LCFS obtained from two consecutive frames in Shot #19993, varying the number of contour lines. The left y-axis represents the IoU, while the right y-axis shows the computation time for both the LCFS and IoU.

The nuclear fusion community has typically measured accuracy by comparing parametric information extractable from the LCFS, such as poloidal angle (Wan et al., 2023). However, these methods are limited to examining only partial aspects of the LCFS shape. This limitation highlights the need for quantitative evaluation metrics that can measure the similarity of the overall LCFS shape.

To address this issue, we experimented with the IoU metric, which is familiar to ML researchers, as a potential method to compare the similarity between the ground truth LCFS and the predicted LCFS. The process begins by drawing $N$ contour lines using Matplotlib (Hunter, 2007). These lines are stored as lists of $(R, Z)$ coordinates, allowing for the identification of closed surfaces by comparing the start and end coordinates of each contour line. The outermost closed surface is selected as the LCFS and converted into a Polygon object of Shapely library to calculate the IoU. We calculated the IoU for each timestep and then reported the mIoU across the entire test data.

While this method of estimating the LCFS and IoU is straightforward, it has limitations that hinder its use as a primary evaluation metric. As illustrated in Figure 6, the IoU calculation is sensitive to the number of contour lines used. Based on this result, we concluded that IoU should be used only as a supplementary evaluation metric to MAE.

