# OpenReview forum: "Data-driven plasma equilibrium forecasting in magnetic fusion tokamak"
_ICLR.cc/2025/Conference — Submitted to ICLR 2025_

### Official Review · Reviewer_jkqE · 2024-11-03

**Soundness:** 3
**Presentation:** 3
**Contribution:** 2
**Rating:** 5
**Confidence:** 3

**Summary:**

This paper addresses the crucial issue of predicting plasma equilibrium in tokamaks, which are essential for advancing nuclear fusion technology. The authors propose a novel method that formulates this prediction task as a video prediction problem, enabling the capture of spatio-temporal dynamics of plasma states. By applying well-established video prediction algorithms, the authors aim to forecast future plasma equilibria based on past states and achieve promising results that contribute to the understanding of plasma instabilities and control.

**Strengths:**

The approach of framing plasma equilibrium prediction as a video prediction task is new in this area, introducing a fresh perspective to the fusion community, which has been largely focused on magnetic field data. This formulation may inspire future research in related fields. The methodology demonstrates a solid understanding of both machine learning techniques and plasma physics, with the integration of a physics-inspired loss function showing potential for improving model predictions. The paper is generally well-organized, providing a clear explanation of the background and motivations behind the research, as well as a structured presentation of methods and results.

**Weaknesses:**

The application of existing video prediction algorithms to the specific context of plasma equilibrium is straightforward and thus may not present sufficient novelty. Further differentiation from previous works is needed to establish the unique contribution of this study.
The selection of the loss function has not been illustrated and validated sufficiently. Equation (2), which should be crucial in this paper, looks strange without an essential explanation.

**Questions:**

1. About the loss function: Why do the authors choose the ell-1 norm, rather than the more frequently used ell-2 norm? In eq. (2), what's the difference between $\Delta^* \psi_{pred}$ and $\Delta^*(\psi_{pred})$?
2. How does the resolution of discretization affect the accuracy of the prediction?
3. Could the authors elaborate on the choice of video prediction algorithms? What specific adaptations were made to ensure compatibility with the plasma equilibrium data?
4. Are there plans to extend this method to include a wider variety of tokamak data, such as temperature profiles or density distributions?

---

> ### Author Response · Authors · 2024-11-21
> **Official Comment by Authors (1/2)**
>
> ## Use of L1 loss instead of L2 loss
> While accurately predicting the overall shape of $\psi$ and $\Delta^{*}\psi$ is important in our problem setting, the most crucial aspect lies in precisely capturing the plasma boundary. It is generally known that L2 loss tends to generate relatively blurry or smooth images and often struggles to represent edges clearly compared to L1 loss.
>
> Based on this reasoning, we hypothesized that L1 loss would be more suitable for better plasma boundary detection. However, we acknowledge the reviewer's valid observation that this hypothesis was not empirically validated in our work. We will investigate the impact of different training loss functions.
>
> ## Explanation of Equation 2
> As the reviewer mentioned, the GS-constraint loss described in Equation 2 plays a crucial role in our work. However, upon reflection, we acknowledge that both the mathematical formulation and its accompanying explanation in the manuscript were not presented with sufficient clarity. To address this, we will refine the notation of the GS-constraint loss with a more detailed explanation in the revised version.
>
> ***(We apologize for mixing symbols due to issues with formula rendering)***
>
> In the current manuscript, the terms in Equation 2 are defined as follows:
> * $\Delta^{*}\psi_{pred}$ refers to the ∆∗ψ directly output by the model.
> * $\Delta^{*}(\psi_{pred})$ denotes the ∆∗ψ obtained by applying the ∆∗ operator (implemented via finite difference) to the model’s ψ output.
>
> While our original intention was to use parentheses to distinguish the application of the $\Delta^{*}$ operator, we now recognize that this notation may have been insufficiently clear to readers. To resolve this ambiguity, we plan to revise the equation to explicitly represent the application of the ∆∗ψ operator as a separate term.
>
> ## Resolution of discretization
> We appreciate the reviewer’s question regarding the "resolution of discretization" and would like to provide clarification based on two possible interpretations.
>
> ### If the question about the spatial resolution of EFIT data
> The EFIT data used in our study corresponds directly to the spatial resolution employed in actual KSTAR operations. Specifically, the KSTAR system is calibrated to a resolution of 65x65, which ensures alignment with its operational parameters. Deviating from this resolution would result in an inconsistency with the real-world application of the system and would render any prediction accuracy comparisons less meaningful. For this reason, the experiments in our study were conducted exclusively at the 65x65 resolution, and we did not consider other resolutions.
>
> ### If the question about using finite differences in GS-constraint loss calculations
> While solutions obtained by the EFIT algorithm numerically satisfy the GS equation, the approximation using finite differences inevitably introduces numerical errors, as demonstrated by the †-marked value (0.04679) in Table 1 of our paper.
>
> Using finite differences is a well-established and reliable method, as noted in reference [4] mentioned by Reviewer gmLx. However, the errors from finite differences mean that the GS equation is not perfectly satisfied. Heavily weighting the loss calculated through finite differences during model training could potentially degrade the prediction accuracies.
>
> To mitigate this issue, we conducted ablation studies to determine the optimal weighting coefficient for the GS-constraint loss term. We tested weights from 1e-4 to 1, although this process was less rigorous than other parts of our experiments. The results indicated that a weight of 0.01 provided the best trade-off between satisfying the GS constraint and maintaining high prediction accuracy. Based on this finding, we used the weight of 0.01 in all subsequent experiments.
>
> [4] Ren et al., “PhyCRNet: Physics-informed convolutional-recurrent network for solving spatiotemporal PDEs”, Computer Methods in Applied Mechanics and Engineering (2022), https://www.sciencedirect.com/science/article/abs/pii/S0045782521006514

---

> ### Author Response · Authors · 2024-11-21
> **Official Comment by Authors (2/2)**
>
> ## Choice of SimVP
> As described in the manuscript, we selected SimVP due to its demonstrated versatility and scalability, which align well with both the current requirements of our study and our future research directions. This choice was particularly motivated by SimVP's flexible architecture, which can seamlessly integrate additional data modalities such as heating information and pressure profiles alongside the EFIT data.
>
> The operational pipeline of SimVP is as follows: input frames are first processed through an encoder that performs spatial downsampling, followed by feature extraction and processing using the Inception module, which employs parallel convolutional layers. Finally, the decoder reconstructs the spatial resolution through upsampling, generating the predicted frames. To ensure smooth downsampling and upsampling operations, the architecture requires even spatial dimensions. Consequently, we cropped one pixel from each dimension of the original 65x65 EFIT data, resulting in 64x64 input resolution. Furthermore, we made simple modifications to the standard model structure, which typically handles one or three-channel videos, to accommodate our two-channel video consisting of $\psi$ and $\Delta^{*}\psi$.
>
> ## Future plan
> We appreciate the reviewer’s interest in the broader direction of our study. We are currently considering both implicit and explicit approaches, which are detailed below.
>
> ### Implicit approach
> The EFIT data employed in our experiments was derived using magnetic field information exclusively. If EFIT data incorporating pressure profile calculations (accounting for a more precise consideration of the p' term in the GS equation) becomes available, our proposed formulation can seamlessly adapt to these enhancements. This would implicitly address the reviewer's concerns regarding temperature profiles and density distributions. We anticipate that such an extension would further align our predictions with real-world tokamak experimental outcomes, strengthening the applicability of our approach.
>
> ### Explicit approach
> Building upon our experimental results (Section 4.4: Input Configuration and Prediction Length), we are actively exploring enhancements to our explicit modeling methodology. While our current approach, which inputting heating information alongside EFIT data, has demonstrated limitations, we propose refining the SimVP architecture to integrate this information more effectively. Specifically, we plan to inject heating information into the intermediate layers (e.g., inception modules) of the model. As a starting point, we intend to employ fundamental techniques such as adaptive instance normalization (AdaIN) [5], modified by adaptive group normalization to better align with the architecture of SimVP.
>
> [5] Huang et al., “Arbitrary Style Transfer in Real-Time With Adaptive Instance Normalization”, ICCV (2017), https://openaccess.thecvf.com/content_iccv_2017/html/Huang_Arbitrary_Style_Transfer_ICCV_2017_paper.html
>
> ## Concluding remark
> We appreciate the reviewer for the continued interest in our work and for highlighting its strengths. As an interdisciplinary team of researchers specializing in both fusion science and machine learning, we have leveraged a solid understanding of both domains to produce the results presented in this paper. We are pleased that the reviewer recognized this aspect of our approach. We remain open to further comments or discussion on any aspect of our work, whether related to the machine learning methodology or its application to nuclear fusion.

---

> ### Author Response · Authors · 2024-11-25
> **Gentle reminder for discussion**
>
> Dear reviewer jkqE,
>
> We are thankful for the reviewer's valuable feedback, which improves the completeness of our paper.
>
> We have already provided detailed responses to the comments, which can be summarized as follows:
> * Revision and additional clarification of the GS-constraint loss (Equation 2).
> * Answers to various questions (e.g., the use of L1 loss, resolution of discretization, and choice of SimVP).
> * Our future plan to incorporate a wider variety of tokamak data.
> Please let us know if you have any further comments or feedback. We will do our best to address them.
>
> Best,
>
> Paper 3629 authors

---

> ### Comment · Reviewer_jkqE · 2024-11-30
>
> Thanks for your feedback. I can now understand these points more clearly. I would like to keep my original score.
> A small suggestion: If you need to capture sharp boundaries, then the ell-1 norm is still not good enough. In the future work, you may consider to use "Total Variation" and its generalizations in image processing area.

---

> > ### Author Response · Authors · 2024-11-30
> > **Thank you for the discussion**
> >
> > Thanks for the reviewer's feedback on our rebuttal.
> >
> > We appreciate the comments and will work to reflect suggestions to improve the paper. If there are any additional comments, please feel free to let us know.

---

### Official Review · Reviewer_T2AC · 2024-11-03

**Soundness:** 2
**Presentation:** 3
**Contribution:** 2
**Rating:** 3
**Confidence:** 4

**Summary:**

The paper introduces a machine learning forecasting technique for applications in tokamak plasma predictions.  The data are spatio-temporal fields characterizing the plasma dynamics.  The method proposed makes use of equilibrium solutions from the past to predict the future states.

**Strengths:**

As for applications, they do propose to apply this to an important problem in physics which is difficult to do forecasting on.  Certainly the characterization of plasmas is an important and challenging data set.

**Weaknesses:**

While I do agree that they are proposing a new method, the comparatives to other methods just aren't up to the level required for ICLR.  Only convLSTM is really compared with for forecasting.  But I would argue there should have been a much richer set of comparatives.  For instance, there is the very simply DMD method for forecasting plasma dynamics:

https://pubs.aip.org/aip/pop/article/27/3/032108/929066

The dynamics they are looking have been shown to be pretty well characterized by a linear model in many application areas.  Using all this technology without comparison to a baseline linear model does not warrant the paper to move forward.  I work in spatio-temporal dynamic systems and just don't find their results compelling enough in terms of advancing the field, nor the results so compelling in terms of the application itself.

And there are other methods that should have been applied beside convLSTM:

ResNet (He et al 2016)
PredRNN (Wang et al 2017)

Also, completely missed for the video prediction task idea is https://www.nature.com/articles/s43588-022-00281-6

Ultimately, I just don't think either the results or their method represent an innovative enough leap (either in the ML/AI architecture proposed or in the advancement of the application) for ICLR.

I cannot recommend the paper moving forward at this point.

**Questions:**

The baselines and comparatives are simply not good enough in my view (see above).  There are other plasma reduced order models for forecasting that are completely ignored.  In addition, the method, while clever, does not signify an significant innovation for ICLR.

---

> ### Author Response · Authors · 2024-11-21
> **Official Comment by Authors**
>
> ## Comparision with linear models and reduced order models
> As the reviewer has highlighted, it is possible to characterize all dynamics using linear models under spatiotemporal constraints. For instance, Taylor expansion is a powerful tool for linear approximation. However, this approach presents practical challenges, such as determining how to approximate specific governing equations and defining the appropriate scope of spatiotemporal constraints. In light of these issues, our work introduces a data-driven approach as an alternative solution, aiming to provide greater flexibility and adaptability in modeling plasma dynamics.
>
> In plasma physics, the Boltzmann equation serves as the fundamental governing equation for describing the spatiotemporal evolution of plasmas. However, solving the Boltzmann equation requires substantial computational resources and time. Consequently, both conventional fusion research and our study rely on the GS equation, which can be regarded as an effectively reduced form of the Boltzmann equation, as an alternative. Furthermore, the spatial distribution of current ($FF^{'}$ term) in the GS equation remains poorly understood and is typically approximated using low-order functions (e.g., first to third-order polynomials). From this perspective, we are already operating within the framework of a reduced plasma dynamics model.
>
> Given these complexities, we acknowledge the importance of considering linear and reduced-order models; however, they require careful consideration from multiple perspectives. Could the reviewer clarify their specific concerns or goals regarding the mention of linear and reduced-order models? If our response does not fully address these concerns, we would welcome further discussion to ensure a comprehensive understanding.
>
> To address the reviewer’s suggestion, we conducted additional experiments using the DMD method to forecast plasma equilibrium states. Details of the methodology and results are provided in our general response. To summarize, the DMD models demonstrated comparable accuracy for short-term predictions (e.g., 50 ms ahead). However, their accuracies declined for longer time horizons, highlighting the limitations of linear models in capturing long-term plasma dynamics. This indicates the need for more sophisticated approaches, such as the one we propose.
>
> ## Additional reference
> We thank the reviewer for pointing out the related work. As our main focus has been on literature spanning nuclear fusion and machine learning, we may have missed this study. The approach in the suggested work, which addresses physical phenomena prediction as a video prediction task, aligns well with our method, and we will make sure to cite it in our manuscript.
>
> The ongoing publication of such research highlights the growing interest in solving physics problems using (video) data-driven methods. We hope that AI/ML conferences such as ICLR will continue to recognize this emerging demand and actively support innovative research in this area.
>
> ## Concluding remark
> The reviewer's expert guidance has helped us strengthen our baseline experiments. Details of the additional baseline experiments are provided in the general response, and we remain open to addressing any further questions or suggestions. Additionally, we welcome comments on our contributions and related content should further discussion be required.
>
> The intersection of machine learning and fusion research offers valuable opportunities for meaningful collaboration. By introducing novel solutions from an ML perspective, we believe this interdisciplinary approach has the potential to drive significant advancements in fusion technology.

---

> ### Author Response · Authors · 2024-11-25
> **Gentle reminder for discussion**
>
> Dear reviewer T2AC,
>
> We are thankful for the reviewer's valuable feedback, which improves the completeness of our paper.
>
> We have already provided detailed responses to the comments, which can be summarized as follows:
> * An in-depth explanation of the contributions and novelty of our study.
> * Comparisons with linear models and reduced-order models, such as DMD.
> * Additional baseline experimental results with models such as Chronos, PredRNN, and PredRNNv2.
> Please let us know if you have any further comments or feedback. We will do our best to address them.
>
> Best,
>
> Paper 3629 authors

---

### Official Review · Reviewer_gmLx · 2024-11-04

**Soundness:** 3
**Presentation:** 3
**Contribution:** 1
**Rating:** 3
**Confidence:** 4

**Summary:**

In this work, the author considers an application for predicting plasma equilibrium, formulated as a video prediction task. The author proposes a physics-informed learning technique built upon an existing CNN-based model. This model can forecast future plasma equilibrium up to 200 ms ahead, outperforming baseline methods.

**Strengths:**

- The author demonstrates an important application on plasma equilibrium forecasting.
- The author generates a new dataset, which can be a contribution if published.

**Weaknesses:**

### Method
- The paper utilizes an existing method, simVP, which should be introduced in more detail.
- The physics-informed loss using finite differences is fairly standard [1].

### Scope
- Overall, I am concerned that the paper may lack sufficient contributions from the ML perspective, making it potentially more suitable for a journal in nuclear fusion.
- Alternatively, if the dataset can be published, it might be more effective to present this work as a benchmark paper comparing existing methods.

[1] Ren, Pu, et al. "PhyCRNet: Physics-informed convolutional-recurrent network for solving spatiotemporal PDEs." *Computer Methods in Applied Mechanics and Engineering* 389 (2022): 114399.

**Questions:**

It seems quite standard to define the task as video prediction. What was the challenge preventing previous works from doing so? It could be helpful to discuss more the contribution from formulating the problem this way.

---

> ### Author Response · Authors · 2024-11-21
> **Official Comment by Authors**
>
> ## Detailed description of the SimVP model
> Anticipating that the primary audience of this paper would be ML researchers, who may be unfamiliar with nuclear fusion, we focused extensively on delivering fusion-related concepts. However, we recognize that this led to insufficient explanation of the video prediction problem and the SimVP model. In our revision, we will enhance the description of these ML components.
>
> ## Additional reference
> We appreciate your introduction of the relevant study that strengthens the soundness of our research. Given the widespread application of physics-informed neural networks and loss functions across diverse domains, we failed to discover this paper in our literature review. We will cite and discuss this paper in our revised manuscript.
>
> ## AI / ML conference vs. Nuclear fusion journal
> As we discussed in our general response, our video prediction approach is novel within the context of nuclear fusion. However, as the reviewer pointed out, this approach may appear relatively straightforward to ML researchers. We believe this observation underscores the significant potential for ML researchers to contribute to fusion science. By presenting our work at an AI/ML conference, we aim to raise awareness of nuclear fusion challenges and encourage further exploration and collaboration.
>
> That said, this contribution is not confined to advancing research in nuclear fusion alone. As we will discuss in the following dataset section, the unique characteristics of fusion data provide valuable opportunities to advance ML techniques. Considering these aspects, we hope the reviewer will evaluate our contributions from an interdisciplinary perspective and support the broader promotion of ML-based nuclear fusion research.
>
> ## Dataset availability
> Unfortunately, due to institutional policies and confidentiality agreements associated with the KSTAR experiments, we are currently unable to release the dataset. We appreciate the reviewer's suggestion regarding dataset publication, as we had not previously considered this possibility. Had we been able to release the dataset, it could have provided significant value to the ML community.
>
> In the KSTAR campaign, fusion experiments are conducted for half of the year, while the remaining time is dedicated to tokamak device upgrades. Consequently, this leads to natural year-to-year variations in the characteristics of experimental data, including EFIT. From an ML perspective, these temporal variations aligns closely with the concepts of continual learning and transfer learning. Indeed, Joung et al. [3] previously demonstrated this by separately training and evaluating models using data from 2017 and 2018. Their work highlights the potential of fusion data as a benchmark for advancing continual and transfer learning techniques, particularly in video prediction tasks.
>
> If circumstances change, we will explore the possibility of dataset release.
>
> [3] Joung et al., “Deep neural network Grad–Shafranov solver constrained with measured magnetic signals”, Nuclear fusion (2020), https://iopscience.iop.org/article/10.1088/1741-4326/ab555f
>
> ## Concluding remark
> We sincerely thank the reviewer again for the constructive feedback. We have addressed the questions regarding our research contributions in the general response and remain open to providing further clarification if needed. We welcome any additional comments or suggestions that could help improve our work further.

---

> ### Author Response · Authors · 2024-11-25
> **Gentle reminder for discussion**
>
> Dear reviewer gmLx,
>
> We are thankful for the reviewer's valuable feedback, which improves the completeness of our paper.
>
> We have already provided detailed responses to the comments, which can be summarized as follows:
> * A thorough explanation of our contributions, novelty, and the challenges addressed compared to previous studies.
> * Clarification regarding the public availability of the dataset.
> * Inclusion of an additional relevant reference.
> Please let us know if you have any further comments or feedback. We will do our best to address them.
>
> Best,
>
> Paper 3629 authors

---

> > ### Comment · Reviewer_gmLx · 2024-12-02
> >
> > - Thanks the author for adding the experiments. In Table 2, it seems SimVP methods are close to the 50ms delay baseline. Does it suggest the 50ms delay baseline is good enough?
> > - when adding the GS loss, how many ms does it take? if it's longer than 50ms, maybe it's not worthy.
> >
> > - Overall the author could explain a bit more on the contribution of formulating predicting plasma equilibrium as a video prediction task. Especailly, the equilibrium (GS) equation 1 is a time-independent equation. Could the author explain it's relationship to the time-dependent dataset? I.e. what is the equilibrium (stationary state in time) that changes along time?

---

> ### Author Response · Authors · 2024-12-04
> **Official Comment by Authors**
>
> Below is our response to the reviewer’s additional questions:
>
> ## Q: It seems SimVP methods are close to the 50ms delay baseline. Does it suggest the 50ms delay baseline is good enough?
>
> In fusion research, different plasma phenomena operate on distinct timescales. Plasma turbulence typically occurs on the microsecond scale, while plasma density and temperature equilibrium changes occur on the millisecond scale, and spatial current distribution changes are observed on the second scale. In this context, a 50 ms delay baseline may be slightly long for capturing plasma density and temperature equilibrium changes, highlighting a limitation of the 50ms baseline. Note that plasma turbulence is outside the scope of our study, while spatial current distribution changes can be reasonably compared using this baseline.
>
> The primary reason we chose the 50 ms baseline is that our EFIT data is computed at 50 ms intervals. Compared to this baseline, our model provides more accurate predictions of plasma dynamics. Significantly, we emphasize that our method outperforms the 50 ms delay baseline even without fully utilizing the control signals such as heating information. We believe that once control signals are properly integrated, our model will be able to predict longer-term plasma evolution with even greater accuracy.
>
> ## Q: When adding the GS loss, how many ms does it take? if it's longer than 50ms, maybe it's not worthy.
>
> The GS loss is only used during model training and does not affect inference time. Furthermore, the overhead introduced by the GS loss during training is negligible. We measured computation time over 1,000 iterations using Intel Xeon Gold 6448Y and NVIDIA H100, with the following average times:
> * Batch size: 1
>   * CPU: 0.2304 ms
>   * GPU: 0.2513 ms
> * Batch size: 128
>   * CPU: 1.0575 ms
>   * GPU: 0.2576 ms
>
> ## Q: Overall the author could explain a bit more on the contribution of formulating predicting plasma equilibrium as a video prediction task. Especailly, the equilibrium (GS) equation 1 is a time-independent equation. Could the author explain it's relationship to the time-dependent dataset? I.e. what is the equilibrium (stationary state in time) that changes along time?
>
> In fusion experiments, plasma undergoes a gradual process of heating, magnetic confinement, maintenance, and eventual deconfinement. As a result, the plasma equilibrium state evolves continuously over time.
>
> At any infinitesimally short moment, the plasma equilibrium can be treated as time-independent, although it is actually time-dependent and evolves continuously over time. Our dataset consists of EFIT computation data captured at 50 ms intervals, with each snapshot satisfying the GS equation at that specific point in time.
>
> Our approach employs video prediction models to capture the temporal evolution of plasma states, while the GS loss ensures that each predicted state satisfies the physical equilibrium constraints defined by the GS equation. This enables us to model both the time-dependent evolution and maintain physical consistency at each step. Through this methodology, we expect to obtain valuable insights into plasma temporal dynamics, a field that has been relatively underexplored in prior studies.

---

### Author Response · Authors · 2024-11-21
**General response (1/3)**

We appreciate all three reviewers for their constructive feedback and valuable comments.

All three reviewers recognized the following strengths of our work:
* The importance of addressing the plasma equilibrium forecasting problem.
* The novel and clever formulation of this problem as a video prediction task.
* A well-organized manuscript that demonstrates solid understanding of both machine learning (ML) and plasma physics.

The primary concerns raised by the reviewers focus on the contributions of our work and the baseline experiments. In this general response, we address these concerns, while more specific points will be discussed in separate comments.

## Contribution of our work
We would like to clarify several key aspects of the novelty and contributions of our study.

The plasma equilibrium reconstruction problem in fusion research has traditionally focused on reconstructing the current equilibrium state based on current magnetic field measurements. Our study takes a fundamentally different approach by forecasting future plasma equilibrium states based on past equilibrium states, without relying on magnetic field data.

The main reason this approach has not been attempted in fusion research is that the Grad-Shafranov (GS) equation, which physically describes plasma equilibrium states, is time-independent. In other words, the GS equation is designed to reconstruct the current equilibrium state from current measurements (i.e., magnetic field data), rather than predicting future equilibrium states. While temporal dynamics of plasma are studied, the corresponding equations are significantly more complex and computationally intensive than solving the GS equation. Consequently, fusion research, including plasma control and analysis, has primarily relied on GS equation-based methodologies, which provide reasonable results over short timeframes.

In this context, our motivation was to address temporal dynamics of plasma beyond the GS equation using a data-driven approach. The main contribution of our work lies in formulating this novel problem and demonstrating its feasibility through initial results. Given the underexplored nature of this direction, we deliberately formulated the problem as an intuitive video prediction task for our experiments.

One may question why we chose to present our results at an AI/ML conference such as ICLR, rather than nuclear fusion journals. This decision was motivated by the fact that our approach is straightforward for ML researchers, as noted by reviewer gmLx. We believe that ML researchers, who are familiar with video prediction tasks and are not constrained by traditional physics approaches, can offer fresh perspectives and potentially bring significant advances to fusion research.

Indeed, the ICLR call for papers explicitly invites submissions on "applications to physical sciences," and the conference has regularly published papers in areas like materials science and bioinformatics, advancing both ML methods and these scientific domains. We believe our work, which applies data-driven approaches to fusion problems, aligns well with the scope of ICLR conference. Unlike these more established domains, fusion research remains relatively unexplored in the AI/ML community. Following the example of previous work such as Spangher et al. [1], we believe ICLR provides an ideal platform for introducing the challenges of nuclear fusion to the ML community.

We acknowledge the reviewers' concerns regarding the limited contributions from an ML perspective. To clarify, our work expands the applications of ML to the underexplored fusion domain, analyzes how existing ML-based video prediction models perform on fusion data, and identifies challenges as well as future research directions. Specifically, we introduced a GS equation-based loss function to improve prediction consistency and demonstrated the current limitations in effectively incorporating tokamak control signals (e.g., heating information) into video prediction models. These findings contribute meaningful techniques and highlight challenges in applying ML to complex physical systems.

In summary, from a fusion research perspective, our key contribution is pioneering the prediction of future plasma equilibrium states and formulating this as a video prediction task. From an ML perspective, we contribute by extending ML applications to an underexplored scientific domain, demonstrating its feasibility, and establishing directions for future research.

[1] Spangher et al., “Position: Opportunities Exist for Machine Learning in Magnetic Fusion Energy”, ICML (2024), https://proceedings.mlr.press/v235/spangher24a.html

---

> ### Author Response · Authors · 2024-11-21
> **General response (2/3)**
>
> ## Baseline experiment
> We acknowledge the lack of baseline experiments in our initial manuscript and appreciate the reviewers’ suggestions for improvement.
>
> As highlighted in the paper, while considerable research exists on reconstructing current plasma equilibrium from magnetic field data, the task of predicting future plasma equilibrium states based solely on past equilibrium data without magnetic field information remains largely unexplored. This novel and unique problem setting presented challenges in identifying suitable baseline models for comparison.
>
> Following the suggestion of reviewer T2AC, we are currently conducting additional baseline experiments, including Dynamic Mode Decomposition (DMD). Preliminary results are presented below, and we will provide updates as further results become available.
>
> ### DMD
> For the DMD experiments, we considered a prediction task using a 200 ms context window ($[x_{t+1}, x_{t+2}, x_{t+3}, x_{t+4}]$) to predict the subsequent 200 ms ($[x_{t+5}, x_{t+6}, x_{t+7}, x_{t+8}]$). The DMD model was fitted on $[x_{t+1}, x_{t+2}, x_{t+3}, x_{t+4}]$, and predictions were generated autoregressively from the last observation: $x_{t+4}$ → $x_{t+5}^{‘}$, $x_{t+5}^{‘}$ → $x_{t+6}^{‘}$, and so on, ultimately producing $[x_{t+5}^{‘}, x_{t+6}^{‘}, x_{t+7}^{‘}, x_{t+8}^{‘}]$ for comparison with ground truth.
>
> However, the EFIT data, represented as 2D images, is not directly compatible with DMD. To address this issue, we implemented two approaches: **1)** flattening the 2D EFIT data into 1D vectors and **2)** performing predictions on a per-pixel basis. For the 2D structure, we explored two configurations: **a)** simultaneous prediction of $\psi$ and $\Delta^{*}\psi$ (i.e., predicting a vector of dimension 64x64x2 = 8,192) and **b)** separate predictions (i.e., predicting two vectors each of dimension 64x64 = 4,096).
>
> ### Chronos
> Chronos [2], a transformer-based pretrained model for univariate time-series forecasting, was evaluated as a baseline due to its strong performance on diverse time-series datasets. We tested Chronos-t5-base, a model with a parameter count comparable to SimVP, and measured its zero-shot prediction performance on fusion data. Final predictions were obtained by averaging the results of 15 repeated samplings for each data point. It is worth noting that Chronos, being a univariate time-series model, performs pixel-level predictions, whereas our method predicts spatial information in a unified manner.
>
> ### Results and analysis
> The results shown in Tables 1 and 2 indicate that while DMD achieves performance comparable to SimVP for predictions 50 ms ahead, its accuracy declines significantly for longer time horizons. From 100 ms onward, the prediction errors increase remarkably, suggesting that linear models such as DMD may suffice for short-term predictions but struggle with longer-term forecasting. Two possible explanations for these limitations are: **1)** the inherently nonlinear nature of plasma dynamics, which cannot be adequately captured by linear models, and **2)** numerical instabilities in the DMD model, particularly due to the combination of autoregressive prediction and high condition numbers.
>
> Chronos, on the other hand, exhibited significant errors from the first timestep prediction, making it unsuitable for plasma equilibrium forecasting. We initially expected that results from Chronos would help analyze the differences between methods that predict spatial information simultaneously and those that do not; however, its accuracy was insufficient to support such an analysis.
>
> [2] Ansari et al., “Chronos: Learning the Language of Time Series”, TMLR (2024), https://openreview.net/forum?id=gerNCVqqtR

---

> ### Author Response · Authors · 2024-11-21
> **General response (2/3)**
>
> **Table 1: Additional qualitative evaluation 1**
> |          **Method**         | **$\psi$ center MAE** | **$\psi$ divertor MAE** | **$\Delta^{*}\psi$ center MAE** | **$\Delta^{*}\psi$ divertor MAE** | **GS constraint** |
> |:---------------------------:|:---------------------:|:-----------------------:|:-------------------------------:|:---------------------------------:|:-----------------:|
> | Baseline (50 ms delay)      |        0.00165        |         0.00161         |             0.01493             |              0.00742              |      0.04679      |
> | SimVP                       |        0.00128        |         0.00124         |             0.01825             |              0.00996              |      0.36242      |
> | SimVP + $\mathcal{L}_{GS}$  |        0.00123        |         0.00125         |             0.02035             |              0.01108              |      0.21476      |
> | SimVP ($\psi$ only)         |        0.00122        |         0.00119         |                -                |                 -                 |         -         |
> | DMD                         |        0.93483        |         0.89436         |             14.4225             |              9.37053              |      5.09277      |
> | DMD ($\psi$ only)           |        0.98264        |         1.26488         |                -                |                 -                 |         -         |
> | DMD ($\Delta^{*}\psi$ only) |           -           |            -            |             9271.37             |              7031.53              |         -         |
> | DMD (pixel level)           |        0.01053        |         0.01028         |             0.03441             |              0.01965              |      0.03163      |
> | Chronos                     |        0.01758        |         0.01756         |             0.06306             |              0.04728              |      47.0440      |
>
> **Table 2: Additional qualitative evaluation 2**
> |         **Method**         | **50 ms** | **100 ms** | **150 ms** | **200 ms** |
> |:--------------------------:|:---------:|:----------:|:----------:|:----------:|
> | Baseline (50 ms delay)     |  0.00167  |   0.00167  |   0.00167  |   0.00168  |
> | SimVP                      |  0.00104  |   0.00121  |   0.00136  |   0.00152  |
> | SimVP + $\mathcal{L}_{GS}$ |  0.00096  |   0.00116  |   0.00132  |   0.00149  |
> | SimVP ($\psi$ only)        |  0.00092  |   0.00112  |   0.00132  |   0.00152  |
> | DMD                        |  0.00212  |   0.00895  |   0.12986  |   3.59782  |
> | DMD ($\psi$ only)          |  0.00186  |   0.00757  |   0.12207  |   3.79866  |
> | DMD (pixel level)          |  0.00133  |   0.01388  |   0.01359  |   0.01333  |
> | Chronos                    |  0.01299  |   0.01612  |   0.01919  |   0.02203  |

---

> ### Author Response · Authors · 2024-11-21
> **General response (3/3)**
>
> ## Revision plan
> We are currently working on revising the manuscript in response to the reviewers’ suggestions, incorporating additional baseline experimental results. We will complete the revision as soon as possible, before the discussion period ends, and we will notify the reviewers once the updates are ready.
>
> Best,
>
> Paper 3629 authors

---

### Author Response · Authors · 2024-11-25
**Paper revision**

Based on the reviewers' feedback, **we have updated our manuscript** to better address their comments and concerns.

The main changes are summarized as follows:
* Enhanced clarity on our contribution and novelty.
* Revised GS-constraint loss equation with additional explanations.
* Added additional baselines, including DMD, Chronos, PredRNN, and PredRNNv2, with detailed descriptions and comparative analysis.
* Incorporated citations to the two papers suggested by the reviewers.

For your convenience, ***all the updated parts in the manuscript are highlighted in pink***.

We kindly invite you to review the revised manuscript for further details and the complete experimental results. We hope these revisions and our responses sufficiently address the reviewers' concerns. Any additional comments or suggestions would be greatly appreciated.

---

### Meta-Review · Area_Chair_Qc4T · 2024-12-19

**Metareview:**

The authors present results on predicting the future plasma state in tokamak experiments from past state, which they claim is an advance beyond previous work on reconstructing current state from magnetic measurements. The reviewers agreed that the baseline comparisons were too limited for an ML venue, while the topic was potentially too far afield and more appropriate for a fusion journal. Additionally, the authors entirely ignore the field of physics-based forward equilibrium evolution. Forward evolutive solvers such as FGE (e.g. _Development of free-boundary equilibrium and transport solvers for simulation and real-time interpretation of tokamak experiments_, F. Carpanese 2021) are sufficiently mature that they can be used for model predictive control and deep reinforcement learning. Yet the authors don't even acknowledge the existence of these tools, let alone compare against them, which makes it unclear what if any advantage these tools would have over existing physics-based simulation approaches. I recommend against acceptance.

**Additional Comments On Reviewer Discussion:**

Unfortunately, the reviewers did not really engage with the authors during the discussion period. Nevertheless, having looked at the authors' rebuttal myself, I don't believe that the changes to the paper were substantial enough to overcome the major objections.

---

### Decision · Program_Chairs · 2025-01-22

Reject